# Evaluating cognitive depth of AI-generated multiple-choice questions with Bloom's Taxonomy

Trang Thi Nguyen[1], Linh Nguyen[2]*, Ha Thi Nguyet Do[2], Huong Thi Thu Nguyen[1], Son Minh Tong[1]

**1** Faculty of Dentistry, Phenikaa University, Hanoi, Vietnam, **2** School of Dentistry, Hanoi medical university, Hanoi, Vietnam

* nguyenlinh.hmu@gmail.com

## Abstract

### Introduction

While LLMs are used to generate medical and dental MCQs, their alignment with Bloom's Taxonomy remains unexplored.

### Materials and Methods

Five widely used LLMs, including ChatGPT-4o (OpenAI), Copilot Pro (Microsoft), Claude Sonnet 4 (Anthropic), Grok 3 (xAI), and DeepSeek R1 (DeepSeek) were evaluated. Each model generated 60 MCQs (total 300) based on content from an oral and maxillofacial anatomy textbook across the five cognitive levels of Bloom's Taxonomy. Two independent investigators assessed each item using a 5-point Likert scale for remembering, understanding, applying, analyzing, and evaluating/creating. Inter-rater reliability was measured using weighted Cohen's kappa. Model performance and inter-model differences were analyzed using the Kruskal–Wallis test.

### Results

Inter-rater reliability was moderate to strong (kappa = 0.74–0.86). Median scores for remembering, understanding, applying, and evaluating/creating were above 4 across all LLMs, while the analyzing level scored a median of 3.5 for ChatGPT-4o and DeepSeek R1. No significant difference was found between models in remembering and understanding levels (p > 0.05). Claude Sonnet 4 outperformed the other models at the applying, analyzing, and evaluating/creating levels (p = 0.01, 0.003, and 0.005, respectively). Within-model analysis showed that only Copilot Pro and Claude Sonnet 4 consistently aligned with Bloom's cognitive levels across all categories. In contrast, ChatGPT-4o, DeepSeek R1, and Grok 3 performed significantly better at the lower cognitive levels (p = 0.00, 0.00, and 0.001, respectively).

**Data availability statement:** All relevant data are within the manuscript and its Supporting Information files.

**Funding:** The author(s) received no specific funding for this work.

**Competing interests:** The authors have declared that no competing interests exist.

## Conclusions

All LLMs performed well at lower cognitive levels, while Claude Sonnet 4 achieved the highest alignment at higher-order levels.

## Introduction

Multiple Choice Questions (MCQs) are widely used in educational systems as a cost-effective way to assess knowledge, comprehension, and problem-solving in large cohorts [1–5], but creating high-quality, cognitively targeted items requires expertise and resources [6,7].

To ensure that MCQs assess a broad range of cognitive skills, educators frequently employ Bloom's Taxonomy—a hierarchical model that classifies learning objectives from basic recall (Remember) to higher-order thinking skills such as application (Apply), analysis (Analyze), and evaluation/creation (Evaluate/Create) [8,9]. This taxonomy helps guide the construction of questions that target specific educational outcomes and cognitive domains [10–13].

In recent years, artificial intelligence (AI) and, more specifically, large language models (LLMs) have demonstrated remarkable capabilities in natural language generation. These models have been used to assist in various educational tasks, such as analyzing data, answering questions, grading, tutoring, and even generating test items [14–17]. With the rapid and ongoing development of LLMs, newer versions often outperform their predecessors in terms of linguistic capacities, contextual understanding, and accuracy [18,19].

While previous studies have explored AI's role in educational applications, including automated MCQ generation [14,20,21], there is a lack of systematic evaluation of how well different LLMs generate questions that align with the cognitive levels outlined in Bloom's Taxonomy. To date, no comparative study has assessed the ability of state-of-the-art LLMs to generate MCQs across Bloom's levels.

This study aims to fill this gap by comparing several current-generation LLMs in their ability to generate MCQs across the spectrum of Bloom's Taxonomy. By analyzing the cognitive depth of AI-generated questions, this research provides insights into the use of LLMs and evaluates their potential as tools for high-quality question generation in medical and dental education.

## Materials and methods

### Study design and duration

This cross-sectional comparative study was conducted from May 25, 2025, to Jun 25, 2025, to evaluate the capacity of LLMs to generate MCQs aligned with Bloom's Taxonomy. The study was approved by the Institutional Ethical Board of the XXX University with the approval number XXX. Informed verbal consent was obtained from all participants, documented in the study records, and witnessed by a member of the research team. A schematic representation of the study workflow is presented in Fig 1.

## Study design

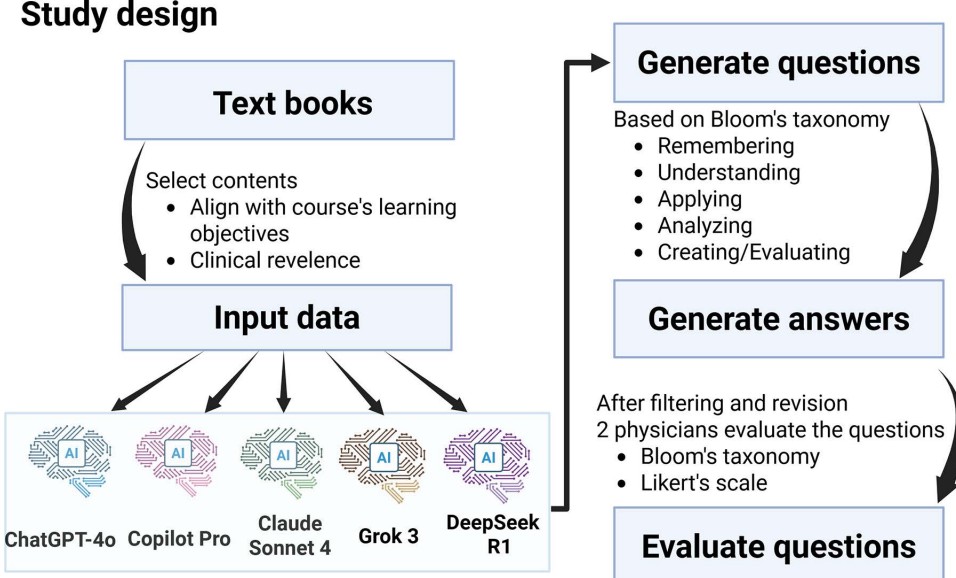

**Fig 1. Flowchart illustrating the study design.**

MCQs and answers were generated using several widely used LLMs, followed by evaluation from two experienced clinicians.

### LLM Selection

Five widely used LLMs were selected based on their public availability, documented widespread use, and technical relevance as of mid-2025. These included: ChatGPT-4o (OpenAI, https://chatgpt.com), released in May 2024; Copilot Pro (Microsoft, powered by GPT-4, https://copilot.microsoft.com), launched in March 2024; Claude Sonnet 4 (Anthropic, https://claude.ai/new), released in May 2025; Grok 3 (xAI, https://grok.com), released in February 2025; and DeepSeek R1 (DeepSeek, https://chat.deepseek.com), introduced in November 2023. Among the selected models, ChatGPT-4o and Copilot Pro required paid subscriptions, whereas Claude Sonnet 4, Grok 3, and DeepSeek R1 were freely accessible. All models were accessed through their respective web interfaces to ensure accessibility, standardization and reproducibility of inputs and outputs.

### Sample size calculation

Sample size was determined using G*Power software (version 3.1) [22], assuming an effect size of 0.1, α level of 0.05, and a statistical power of 0.95. The minimum required sample size per group was calculated to be 38. To increase analytical robustness, 60 MCQs were generated per LLM, resulting in a total of 300 items.

### Source material and prompt design

Input content was derived from the Textbook of Oral and Maxillofacial Anatomy, published by Phenikaa University, Hanoi, Vietnam, in January 2025. Six chapters were selected based on their alignment with curricular learning objectives and clinical relevance:

- The skeletal anatomy of the head and neck

- Musculature of the head and neck

- Temporomandibular joint anatomy

- Vascular anatomy of the head and neck

- Neuroanatomy of the head and neck

- Lymphatic anatomy of the head and neck

Each LLM was prompted independently using a standardized instruction:

"Generate 10 dental board-style multiple-choice questions based on the anatomy of the head and neck as covered in the uploaded files. The number of questions should be distributed across Bloom's Taxonomy as follows: 3 at the Remembering level, 2 at the Understanding level, 3 at the Applying level, 1 at the Analyzing level, and 1 at the Evaluating/Creating level. Each question must include four clearly written and distinct answer choices (A–D), with one correct answer and three plausible distractors. For each question, provide a brief explanation indicating why the correct answer is correct and why each incorrect option is incorrect."

Ten MCQs were generated for each chapter, resulting in 60 MCQs per LLM. Examples of generated questions by ChatGPT-4o in each cognitive level of Bloom's Taxonomy were presented in Table 1.

## Question evaluation

Each question was independently evaluated by two board-certified dental practitioners with over five years of clinical experience and academic appointments as lecturers in head and neck anatomy, each having at least three years of teaching experience. The evaluation was based on the five levels of Bloom's Taxonomy, using a 5-point Likert scale. Final

**Table 1. Example of generated questions and their respective responses by ChatGPT-4o.**

| Questions and answers | Bloom's level/<br>Likert score |
|---|---|
| **Question**<br>Which of the following bones is not part of the midface?<br>A. Maxilla<br>B. Zygomatic bone<br>C. Nasal bone<br>D. Mandible<br>**Correct Answer:** D. Mandible<br>**Explanation:**<br>D is correct because the mandible belongs to the lower face, not the midface. A, B, and C are all part of the midfacial skeleton. | Remembering/<br>Likert score = 5 |
| **Question**<br>You're designing a surgical approach to treat a medially displaced zygomatic fracture. Which structure should you prioritize to restore proper alignment and cosmetic contour?<br>A. Mandibular condyle<br>B. Orbital floor<br>C. Zygomaticofrontal suture<br>D. Infraorbital nerve<br>**Correct answer:** C. Zygomaticofrontal suture<br>**Explanation:**<br>C is key for restoring lateral cheek prominence and alignment of zygoma.<br>A is irrelevant — lower face.<br>B is important but secondary to positioning the zygoma.<br>D is a concern for sensation, not structural restoration. | Evaluating/Creating/<br>Likert score = 3 |

scores were calculated as the average of the two individual ratings. Table 2 summarizes the scoring scheme used for each level of Bloom's Taxonomy.

## Statistical analysis

Statistical analyses were performed using SPSS software (version 23.0; IBM, Armonk, NY). Inter-rater reliability for each of the five evaluation criteria was assessed using weighted Cohen's kappa. The performance of each LLM was reported as the median and interquartile range (IQR). As the data were ordinal, inter- and intra-model comparisons were conducted using the Kruskal–Wallis test, followed by Bonferroni-corrected post hoc tests. Graphs were generated using Python version 3.12.8 (https://www.python.org).

## Results

### Inter-rater reliability

To assess the consistency of two evaluators scoring AI-generated MCQs, we computed weighted Cohen's kappa coefficients ($\kappa$) for each level of Bloom's Taxonomy. Weighted Cohen's kappa measures agreement for ordinal data, with values ranging from 0 (no agreement) to 1 (perfect agreement) [23]. The results showed moderate to strong agreement between evaluators: Remembering ($\kappa = 0.74$), Understanding ($\kappa = 0.77$), Applying ($\kappa = 0.86$), Analyzing ($\kappa = 0.81$), and Evaluating/Creating ($\kappa = 0.78$).

### Comparison among five LLMs

Table 3 summarizes the median and interquartile range (IQR) of Likert ratings for each LLM across the five levels of Bloom's Taxonomy.

Values in parentheses represent the interquartile range (IQR).

Overall, scores clustered around 4 and 5, indicating that all models generally demonstrated adequate to strong performance in generating cognitively aligned multiple-choice questions (S1 Fig).

Ratings predominantly clustered at scores of 4 and 5, indicating that all models generally demonstrated adequate to strong performance in generating multiple-choice questions aligned with Bloom's cognitive levels.

To further visualize performance patterns across cognitive domains, a heatmap of the mean ± standard deviation (SD) Likert scores is presented in Fig 2, illustrating how each model performed across the taxonomy levels.

All models achieved mean scores above 4, indicating generally strong performance. Among them, Claude Sonnet 4 demonstrated the most consistent and highest scores across all cognitive levels.

Statistical comparisons revealed no significant differences among the models at the Remembering and Understanding levels ($p > 0.05$, $\eta^2 = 0.00$). However, notable differences emerged at higher cognitive levels. At the Applying level, Claude Sonnet 4 (rank = 61.44) significantly outperformed Grok 3 (rank = 37.53, $p = 0.033$) and DeepSeek R1 (rank = 35.56, $p = 0.015$) ($\eta^2 = 0.11$). Similarly, at the Analyzing level, Claude Sonnet 4 again demonstrated superior performance compared to ChatGPT-4o ($p = 0.011$) and DeepSeek R1 ($p = 0.015$) ($\eta^2 = 0.47$). At the Evaluating/

**Table 2. Scoring scheme for evaluating the five Bloom's cognitive levels using a 5-point Likert scale.**

| 1 | 2 | 3 | 4 | 5 |
|---|---|---|---|---|
| **No alignment** | **Weak alignment** | **Moderate alignment** | **Good alignment** | **Excellent alignment** |
| Does not match the intended Bloom's level or cognitive demand. | Poorly reflects the intended level; too simple or too complex. | Partially reflects the level; some overlap with other levels or lacks precision. | Mostly appropriate for the level, with minor issues in depth or clarity. | Strongly reflects the intended Bloom's level; clear and cognitively appropriate. |

**Table 3. The median and interquartile range (IQR) of the score of each LLM across different Bloom's cognitive levels.**

| Levels | LLMs | | | | |
|---|---|---|---|---|---|
| | ChatGPT-4o | Copilot Pro | Claude Sonnet 4 | Grok 3 | DeepSeek R1 |
| Remembering | 5.0 (5.0-5.0) | 5.0 (4.87-5.0) | 5.0 (5.0-5.0) | 5.0 (5.0-5.0) | 5.0 (5.0-5.0) |
| Understanding | 4.5 (4.0-5.0) | 4.5 (3.25-5.0) | 5.0 (4.0-5.0) | 4.5 (4.0-5.0) | 5.0 (4.5-5.0) |
| Applying | 4.0 (4.0-5.0) | 5.0 (4.0-5.0) | 5.0 (4.75-5.0) | 4.0 (3.0-5.0) | 4.0 (3.5-4.5) |
| Analyzing | 3.5 (2.75-4.0) | 5.0 (3.75-5.0) | 4.0 (4.0-4.0) | 4.0 (3.75-5.0) | 3.5 (2.5-4.125) |
| Evaluating/ Creating | 4.0 (3.0-4.0) | 4.0 (4.0-5.0) | 5.0 (4.75-5.0) | 5.0 (4.375-5.0) | 4.0 (3.375-4.5) |

# Mean Likert Scores per Bloom's cognitive level across LLMs

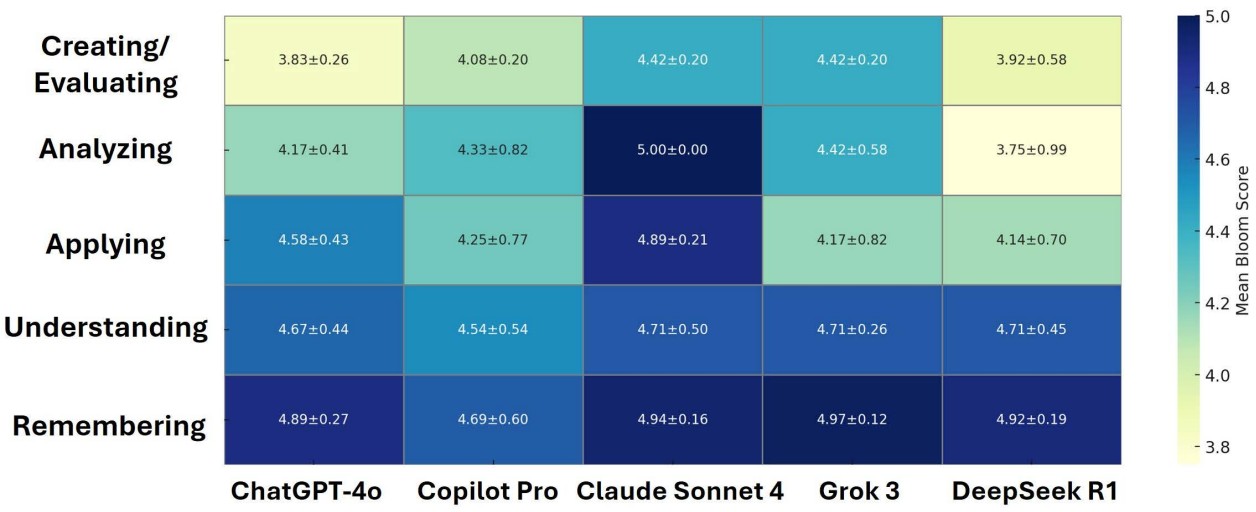

Fig 2. Heatmap of mean ± standard deviation (SD) Likert scores for Bloom's cognitive levels across five LLMs.

Creating level, both Claude Sonnet 4 (rank = 22.58) and Grok 3 (rank = 21.42) significantly outperformed ChatGPT-4o (rank = 7.67; $p < 0.05$) (($\eta^2 = 0.44$) (Fig 3).

Boxplots represent the median (horizontal line), interquartile range (IQR), and mean (black dot) of Likert scores for AI-generated MCQs, categorized by LLMs. A) Remembering, B) Understanding, C) Applying 4, D) Analyzing, E) Evaluating/Creating. No significant differences were found at the Remembering and Understanding levels ($p > 0.05$). At higher levels, Claude Sonnet 4 showed superior performance at Applying and Analyzing ($p < 0.05$), and together with Grok 3 at Evaluating/Creating ($p < 0.05$). Statistical significance: *$p < 0.05$; Kruskal–Wallis test with Bonferroni correction.

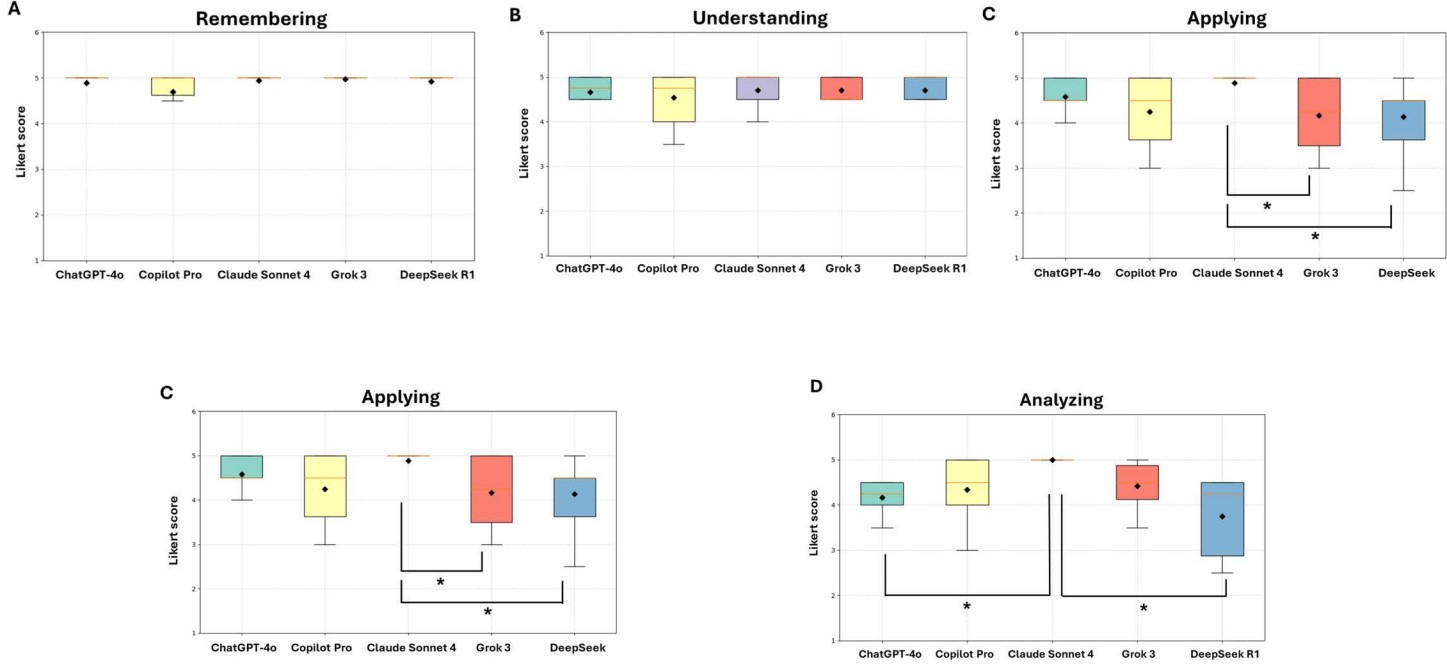

**Fig 3. Distribution of Likert scores across Bloom's cognitive levels.**

## Performance of LLMs across cognitive levels

Copilot Pro and Claude Sonnet 4 showed no significant variation in question quality across cognitive levels ($p > 0.05$, $\eta^2 = 0.01$). In contrast, ChatGPT-4o, Grok 3, and DeepSeek R1 exhibited significant differences ($p < 0.01$). For ChatGPT-4o, Remembering items (mean rank = 41.42) scores higher than Analyzing (rank = 12.67) and Evaluating/Creating (rank = 16.00) ($p < 0.01$, $\eta^2 = 0.31$). Grok 3 also performed better at Remembering level than Applying level ($p < 0.01$, $\eta^2 = 0.25$). In DeepSeek R1, lower-order levels (Remembering, Understanding) scored significantly higher than higher-order ones (Applying, Analyzing, Evaluating/Creating) ($p \le 0.05$, $\eta^2 = 0.50$) (Fig 4).

Boxplots represent the median (horizontal line), interquartile range (IQR), and mean (black dot) of Likert scores for AI-generated MCQs, categorized by Bloom's cognitive levels. A) ChatGPT-4o, B) Copilot Pro, C) Claude Sonnet 4, D) Grok 3, E) DeepSeek R1. Copilot Pro and Claude Sonnet 4 showed consistent performance across all levels ($p > 0.05$). ChatGPT-4o, Grok 3, and DeepSeek R1 demonstrated significant variation ($p < 0.01$), with higher quality at lower cognitive levels compared to higher-order levels. Statistical significance: *$p < 0.05$, **$p < 0.001$; Kruskal–Wallis test with Bonferroni correction.

## Discussion

This study provides a detailed evaluation of five LLMs—ChatGPT-4o, Copilot Pro, Claude Sonnet 4, Grok 3, and DeepSeek R1—in generating multiple-choice questions aligned with Bloom's Taxonomy, a widely used framework for categorizing cognitive skills in educational settings. To our knowledge, no prior study has systematically investigated the alignment of LLM-generated questions with Bloom's cognitive taxonomy. By including a range of LLMs, this study offers a comprehensive view of their potential and applicability in real-world educational tasks.

The results revealed that while all models excel at generating MCQs for foundational cognitive levels (Remembering and Understanding), their performance varies significantly at higher-order levels (Applying, Analyzing, Evaluating/

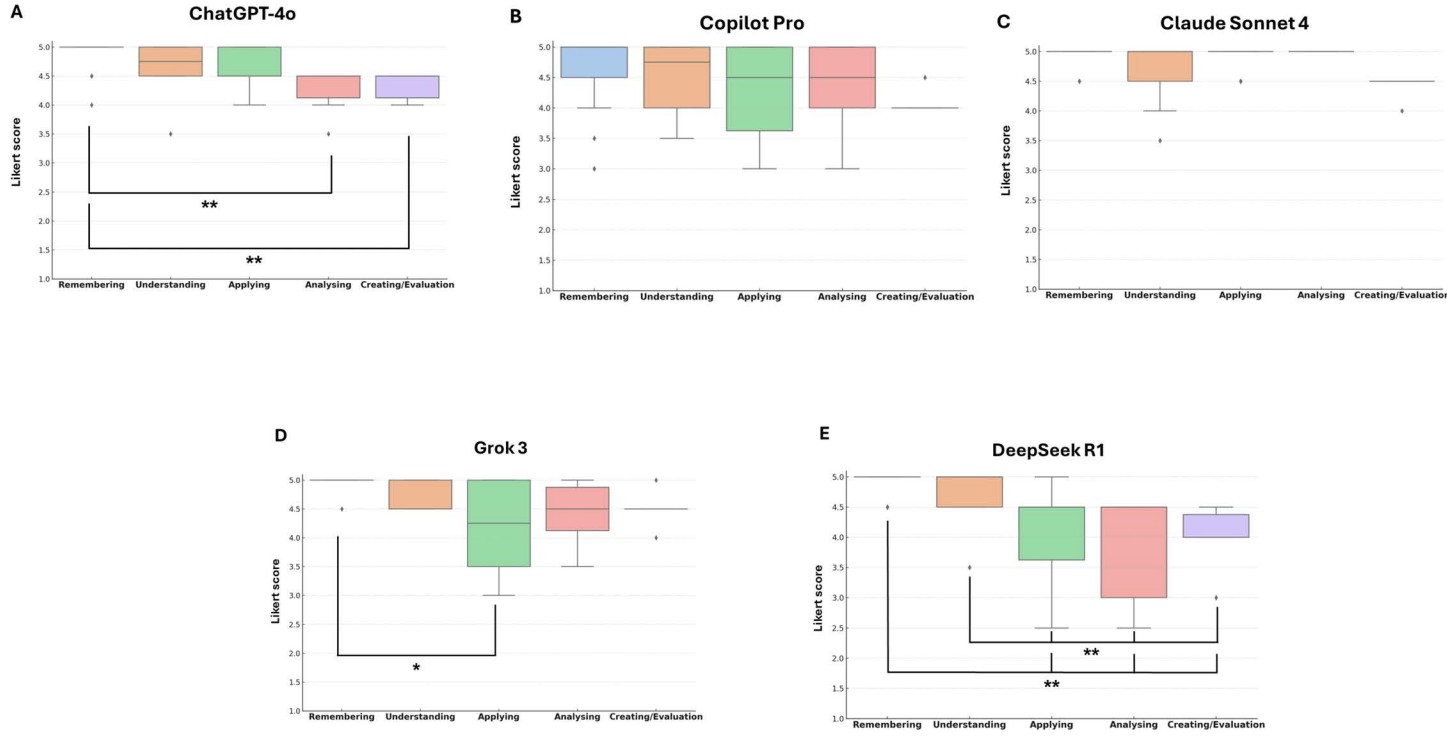

**Fig 4. Distribution of Likert scores across Bloom's cognitive levels for each LLM.**

Creating). Notably, Claude Sonnet 4 achieved higher alignment at higher levels than ChatGPT-4o, Grok 3 and DeepSeek, highlighting variability in LLM capabilities for tasks requiring abstraction and synthesis. Although no previous studies have directly compared LLMs' capabilities to generate MCQs aligned with Bloom's Taxonomy, several investigations have demonstrated that LLMs can produce dental and medical questions with high levels of accuracy, relevance, and complexity [14,20]. Previous studies comparing the question-answering performance of ChatGPT and Claude have reported mixed results. For example, one study evaluating pediatric dentistry questions found that ChatGPT-4 outperformed Claude [24]. In contrast, a separate investigation focusing on colorectal cancer-related queries showed that Claude achieved higher accuracy (82.67%) compared to ChatGPT-4 Turbo (78.44%) and Bard (Gemini) (70%) [25]. These discrepancies may be attributed to differences in the specific models evaluated, as well as the rapid pace of AI development, which has led to the release of newer models with enhanced reasoning capabilities.

When comparing scores across cognitive levels within each model, Copilot Pro and Claude Sonnet 4 demonstrated similar capabilities in generating questions across cognitive levels. In contrast, ChatGPT-4o and DeepSeek R1 performed better at lower cognitive levels than at higher ones. This observation is consistent with findings by Alex KK Law et al. [21], who reported that although ChatGPT-4o was able to generate questions of comparable quality to those produced by human experts, the difficulty index of its questions was significantly lower. Similarly, research assessing LLM performance on the Japanese medical licensing exam found that accuracy rates were higher for easier questions compared to more challenging ones [26]. However, it is important to note that while Alex KK Law et al concluded that LLMs mainly generate questions at lower cognitive levels, their study did not specifically instruct the models to produce questions aligned with Bloom's Taxonomy. In our evaluation, even at higher cognitive levels, models such as Claude, Grok 3, and Copilot Pro were able to generate questions with relatively high scores (above 4), with a narrow IQR (0–1). These findings underscore the importance of selecting LLMs based on the specific cognitive demands of educational tasks, particularly in healthcare

education, where higher-order reasoning skills are critical [27]. Future research should explore advanced fine-tuning methods, curated datasets, and prompt engineering techniques, such as chain-of-thought prompting, to further address these performance gaps [28,29].

Several factors may contribute to variability in LLM performance when generating multiple-choice questions aligned with Bloom's Taxonomy. These include differences in architecture, training data, and fine-tuning strategies, which influence effectiveness across cognitive levels from Remembering to Evaluating/Creating. LLMs rely on transformer architectures using self-attention to process complex inputs [30]. Architectural choices—such as layer depth, attention mechanisms, or Mixture of Experts (MoE)—can influence cognitive performance. Optimized attention or well-configured MoE may support higher-order tasks, while decoder-only models often favor fluency over deep reasoning [31,32]. Training data quality and diversity also matter. Pretraining on abstract or technical content may enhance reasoning and synthesis abilities [33,34]. Fine-tuning methods—such as Low-Rank Adaptation (LoRA), reinforcement learning with human feedback (RLHF), or Retrieval-Augmented Generation (RAG)—can further improve performance by aligning models with specific educational goals [35–37].

The observed performance differences have implications for educational applications, particularly in fields like healthcare, where higher-order cognitive engagement is critical for long-term knowledge retention and clinical problem-solving [27,29]. For instance, Claude Sonnet 4's ability to generate high-quality MCQs at the higher cognitive levels suggests it can support educators in designing assessments that foster critical thinking and innovation, such as creating novel treatment plans or evaluating competing clinical approaches. Similarly, Grok 3's strong performance at Evaluating/Creating level indicates its potential for generating questions that require students to synthesize information, and problem-solving, the skill essential for evidence-based practice. In contrast, ChatGPT-4o's lower performance at Analyzing and DeepSeek R1's moderate scores at higher levels suggest limitations in their ability to handle tasks requiring deeper reasoning or concepts with complexities. Automatically generated MCQs also enable medical students to reinforce their knowledge efficiently by providing virtually unlimited practice questions in a short period. In addition, these tools can tailor both the difficulty and cognitive level of the questions to match learners' needs and stages of training. For example, undergraduate students may benefit from questions with less emphasis on case-based scenarios or complex reasoning, while those preparing for national board examinations or postgraduate assessments often require more advanced, higher-order questions [38]. In practice, educators may selectively use Claude Sonnet 4 for developing advanced assessments targeting higher-order cognition, whereas ChatGPT-4o may be more suitable for foundational knowledge testing. While LLMs can markedly shorten the time needed to generate multiple-choice questions, some level of expert review remains necessary to refine item clarity, ensure accuracy, and verify cognitive alignment. The overall time required for MCQ development may therefore shift from question creation to quality assurance. Additionally, effective utilization of LLMs requires a basic understanding of prompt formulation to elicit high-quality outputs. Although proficiency in prompt engineering may initially represent a barrier to adoption, the increasing accessibility of LLM interfaces and guided educational tools is expected to minimize this challenge over time.

Our study strength was that we compared both free and paid LLMs, providing a broad and balanced perspective on model performance across a rapidly evolving AI landscape. Despite subscription models offering greater usage limits and faster response speeds, certain free LLMs, particularly Claude Sonnet 4 and Grok 3 demonstrated equally strong or even superior performance at advanced cognitive levels (Table 4).

Furthermore, by identifying models' strengths and limitations at each cognitive level, this study establishes a valuable baseline for future work in AI-assisted instructional design, adaptive learning systems, and multimodal assessment tools. Despite its contributions, this study has several limitations that should be acknowledged. While two independent evaluators achieved strong inter-rater reliability ($\kappa = 0.74–0.86$), the process remained subjective, with potential variation in interpreting Bloom's levels and question quality. The evaluation was also limited to MCQs generated from oral and maxillofacial anatomy content, which may restrict the generalizability of the findings to other medical or dental disciplines. In

**Table 4. Comparative Performance of Paid and Free Large Language Models (LLMs) in Generating MCQs Aligned with Bloom's Taxonomy.**

| Access Type | Model | Overall Performance (by Bloom's Level) |
|---|---|---|
| Paid LLMs (Subscription) | ChatGPT-4o (OpenAI) | Moderate – best for foundational questions |
| | Copilot Pro (Microsoft) | High – reliable across cognitive levels |
| Free LLMs | Claude Sonnet 4 (Anthropic) | Highest – excellent for higher-order cognition |
| | Grok 3 (xAI) | High – good for synthesis and problem-solving |
| | DeepSeek R1 (DeepSeek) | Moderate – suitable for foundational questions |

addition, the structure and wording of the prompts used to guide LLM outputs may have influenced the quality and cognitive alignment of the generated questions, introducing potential bias. Moreover, while our study focused specifically on alignment with Bloom's Taxonomy, without assessing aspects such as clarity, clinical relevance, or suitability for dental exams, future research should address these dimensions and explore their impact on student learning outcomes. Finally, the analysis was limited to text-based MCQs, restricting its relevance to multimodal assessments. As advancements in medical imaging continue to address challenges in data generation and quality enhancement, future research exploring the potential of LLMs could offer deeper insights into their capabilities and transformative applications in this domain [39].

## Conclusions

In conclusion, this study highlights significant variation among LLMs in their ability to generate MCQs aligned with Bloom's Taxonomy, with Claude Sonnet 4 demonstrating the strongest ability to generate higher-order questions, while all evaluated LLMs performed well at lower cognitive levels. The findings offer insights for educators seeking to integrate AI into curriculum design and for developers aiming to enhance LLM capabilities for educational purposes. Continued research and model improvement may further enhance the effectiveness of LLMs in supporting critical thinking and deeper learning within healthcare education.

## Supporting information

**S1 Fig. Distribution of Likert ratings assigned by both evaluators across all large language models.**
(TIFF)

**S1 File. Raw expert scoring data for MCQs generated by each LLM, used for all analyses reported in the manuscript.**
(XLSX)

**S2 File. Source code.**
(HTML)

## Acknowledgments

We are grateful to our colleagues for their valuable scientific discussions and contributions throughout the course of this research.

## Author contributions

**Conceptualization:** Trang Thi Nguyen, Linh Nguyen, Son Minh Tong.

**Data curation:** Trang Thi Nguyen, Linh Nguyen.

**Formal analysis:** Trang Thi Nguyen.

**Funding acquisition:** Trang Thi Nguyen.

**Investigation:** Trang Thi Nguyen, Linh Nguyen, Ha Thi Nguyet Do, Huong Thi Thu Nguyen.

**Methodology:** Trang Thi Nguyen, Linh Nguyen, Huong Thi Thu Nguyen, Son Minh Tong.

**Project administration:** Son Minh Tong.

**Resources:** Trang Thi Nguyen, Linh Nguyen, Huong Thi Thu Nguyen.

**Software:** Linh Nguyen.

**Supervision:** Huong Thi Thu Nguyen, Son Minh Tong.

**Validation:** Trang Thi Nguyen, Linh Nguyen.

**Visualization:** Linh Nguyen, Ha Thi Nguyet Do.

**Writing – original draft:** Trang Thi Nguyen, Linh Nguyen.

**Writing – review & editing:** Trang Thi Nguyen, Linh Nguyen, Ha Thi Nguyet Do.

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
