## [Decision Letter · Decision Letter 0]

15 Oct 2025

Dear Dr. Linh,

We look forward to receiving your revised manuscript.

Kind regards,

Musa Adekunle Ayanwale, Ph.D

Academic Editor

PLOS ONE

Journal Requirements:

3. Please note that PLOS One has specific guidelines on code sharing for submissions in which author-generated code underpins the findings in the manuscript. In these cases, we expect all author-generated code to be made available without restrictions upon publication of the work. Please review our guidelines at https://journals.plos.org/plosone/s/materials-and-software-sharing#loc-sharing-code and ensure that your code is shared in a way that follows best practice and facilitates reproducibility and reuse

4. Please ensure that you include a title page within your main document. You should list all authors and all affiliations as per our author instructions and clearly indicate the corresponding author.

5. We notice that your supplementary figures are uploaded with the file type 'Figure'. Please amend the file type to 'Supporting Information'. Please ensure that each Supporting Information file has a legend listed in the manuscript after the references list.

Reviewers' comments:

Reviewer's Responses to Questions

**Comments to the Author**

1. Is the manuscript technically sound, and do the data support the conclusions?

Reviewer #1: Yes

Reviewer #2: Yes

2. Has the statistical analysis been performed appropriately and rigorously?

Reviewer #1: Yes

Reviewer #2: Yes

3. Have the authors made all data underlying the findings in their manuscript fully available?

Reviewer #1: Yes

Reviewer #2: Yes

4. Is the manuscript presented in an intelligible fashion and written in standard English?

Reviewer #1: Yes

Reviewer #2: Yes

Reviewer #1: It is a great pleasure to have reviewed this paper, “Evaluating Cognitive Depth in AI-Generated Multiple-Choice Questions: A Comparative Study of Modern Large Language Models Using Bloom’s Taxonomy”. The study has numerous merits, but for this review, I will unravel some grey areas that, after implementing the correction, will further strengthen the quality of the paper.

Title and Abstract

Title (L1–3): The title is informative and clearly indicates scope, methods, and framework. However, it is a little long. You might shorten to: “Evaluating Cognitive Depth of AI-Generated Multiple-Choice Questions with Bloom’s Taxonomy”. The “Short Title” already captures this succinctly.

Abstract (L4–30): The abstract is structured but slightly repetitive. The “Introduction” sentence could be condensed into one line (“While LLMs are used to generate medical and dental MCQs, their alignment with Bloom’s Taxonomy remains unexplored.”). The “Materials and Methods” section is clear, but you should specify that each model generated 60 MCQs (total 300) upfront for transparency. In the “Results,” provide exact p-values where significant, not just p < 0.05. In “Conclusions,” avoid subjective phrasing (“superior performance”) and instead state “Claude Sonnet 4 achieved the highest alignment at higher-order levels.”

Introduction

Opening (L31–38): The rationale for MCQs is well stated, but the phrasing is verbose. Consider shortening: “MCQs provide a cost-effective way to assess knowledge, comprehension, and problem-solving in large cohorts, but creating high-quality, cognitively targeted items requires expertise and resources.”

Bloom’s Taxonomy (L39–43): Good explanation. Ensure consistent naming (“Analyze,” “Evaluate/Create”) throughout the manuscript. Consider referencing updated frameworks (e.g., Anderson & Krathwohl’s 2001 revision).

AI and LLM background (L44–49): Strong context, but some citations are older (e.g., GPT-3 work). Since you are submitting in 2025, reference more recent reviews (e.g., Küchemann et al., 2025).

Research gap (L50–57): Well identified, but should emphasize novelty: “To date, no comparative evaluation of state-of-the-art LLMs has assessed their ability to generate MCQs across Bloom’s levels.”

Methods

Design (L60–67): Clear statement of cross-sectional comparative design. Consider explicitly noting IRB exemption if no human subjects were involved.

LLM Selection (L68–76): Strong justification, listing release dates and subscription status. Add citations or URLs to each model’s technical report for transparency.

Sample Size (L77–81): You correctly reference G*Power. Specify the test family (e.g., Kruskal–Wallis) and effect size convention. Good decision to use 60 items per LLM.

Source Material and Prompt (L82–99): Very detailed, which is a strength. However, Table 1 is overly long—condense to 1–2 illustrative examples. Avoid full explanations that overwhelm the reader.

Evaluation (L105–115): Well structured with two independent raters. Provide rater demographics (teaching years, academic level) for credibility.

Statistics (L117–123): You describe use of SPSS and Python. Good practice. Re-word sentence at L121–122 for clarity: “Inter-model comparisons were conducted using the Kruskal–Wallis test, followed by Bonferroni-corrected post hoc tests.”

Results

Inter-rater reliability (L126–133): Good reporting of weighted kappa values. Consider citing Yilmaz & Demirhan (2023) for weighted kappa interpretation.

Model comparison (L135–139): Table 3 is comprehensive. Ensure alignment in table formatting (some rows misaligned). Report effect sizes (η² or rank-biserial) alongside p-values.

Figures (L148–179): Figures 2–4 are useful but captions are too brief. Add interpretive context (e.g., “Claude Sonnet 4 scored highest at Applying, Analyzing, and Evaluating/Creating levels”).

Text results (L166–174): The within-model analysis is strong, but rephrase sentences for conciseness. Example: “For ChatGPT-4o, Remembering items scored higher than Analyzing and Evaluating/Creating (p < 0.01).”

Discussion

Strengths of study (L180–186): Clear articulation of novelty. Good acknowledgment that this is the first systematic LLM-Bloom evaluation.

Interpretation (L187–218): Insightful comparison, but avoid over-interpreting. For example, stating “Claude Sonnet 4 consistently outperformed…” should be softened: “Claude Sonnet 4 achieved higher alignment at higher levels.”

Integration with prior work (L193–200): Nicely cites medical/dental MCQ research. Ensure consistent referencing style (some journal abbreviations missing).

Mechanisms (L219–229): Well-reasoned discussion on architecture, training data, and fine-tuning. Ensure references to Vaswani (2017), Fedus (2022), and Kaplan (2020) are up to date.

Educational implications (L231–247): Strong discussion of healthcare education implications. Add practical takeaway: “Educators may selectively use Claude or Copilot for advanced assessments, while ChatGPT-4o may be more suitable for foundational testing.”

Limitations (L252–260): Limitations are acknowledged (subjectivity of raters, Bloom focus only, exclusion of multimodal tasks). Strengthen by explicitly noting: (1) only dental anatomy domain, (2) prompt design may bias outputs.

Conclusions

Concluding sentence (L263–269): Strong, but rephrase for precision: “Claude Sonnet 4 demonstrated the highest ability to generate higher-order questions, while all LLMs performed well at lower levels.” Avoid aspirational phrases (“fully realized”).

References

Strengths: Includes foundational MCQ literature and recent 2023–2025 LLM evaluations.

Weaknesses: Some inconsistencies (missing italics for journal names, inconsistent use of initials). Ensure uniform adherence to PLOS ONE reference style. Verify DOI formatting (should be lowercase “doi:” not capitalized).

Overall Assessment

Strengths:

Novel and timely evaluation of LLMs in education.

Strong methodological rigor (sample size, inter-rater reliability, nonparametric statistics).

Practical implications for dental/medical education.

Weaknesses:

Verbosity in Abstract, Introduction, and Methods.

Overly detailed Table 1 examples.

Need for clearer limitations (domain, prompt, subjective scoring).

Figures and tables require more explanatory captions.

References need style consistency.

Recommendations:

1. Shorten Abstract and Introduction for concision.

2. Reduce Table 1 to 1–2 examples; move others to Supplementary.

3. Add effect sizes to results.

4. Rephrase strong claims in Discussion and Conclusion.

5. Expand Limitations section explicitly.

6. Revise reference formatting per PLOS ONE guidelines.

Reviewer #2: This is an interesting research which has the potential of having direct impact on professional teaching similar course. I would like the authors to address of the thoughts I had as I read the manuscript in the hope to making it better for readers.

Provide some citation to back the statement in line 50-52

It will be more helpful to have the figures at the appropriate location they are mentioned in the manuscript instead of after reference. Not seeing those figures break flow of reading.

In table 3, I am not sure what the values in the parenthesis represents. Some clarification would be helpful to readers.

I was hoping so see some kind of quick highlight table showing the performance based on free and paid LLMs as stated in line 248. Is there any takeaway from that?

Can you provide some justification for the use of Kruskal-Wallis? Line 121

I am wondering if the use of LLM will reduce the usual time used for developing MCQ? Of course it would potentially reduce time it takes to write but what about need for editing? Would there be need for proficiency in prompt engineering? Could that be a roadblock for potential use?

**Do you want your identity to be public for this peer review?** For information about this choice, including consent withdrawal, please see our Privacy Policy

Reviewer #1: **Yes:** Oluwaseyi Aina Gbolade Opesemowo

Reviewer #2: **Yes:** Daniel Oyeniran

---

## [Author Response · Author response to Decision Letter 1]

11 Nov 2025

Thank you very much for your time and for the valuable, constructive feedback provided on our manuscript. The following files have been uploaded for your kind review:

- Response to reviewers (addressing all comments in detail)

- Revised manuscript with tracked changes

- Clean revised manuscript

Please kindly let us know if any additional materials or clarifications are required for re-review. We are sincerely grateful for the editors’ and reviewers’ time and thoughtful consideration, and we look forward to your response.

With kind regards,

Corresponding author,

on behalf of all authors

---

## [Decision Letter · Decision Letter 1]

6 Jan 2026

Title: Evaluating Cognitive Depth of AI-Generated Multiple-Choice Questions with Bloom’s Taxonomy

PONE-D-25-36938R1

Dear Dr. Nguyen,

We’re pleased to inform you that your manuscript has been judged scientifically suitable for publication and will be formally accepted for publication once it meets all outstanding technical requirements.

Kind regards,

Musa Adekunle Ayanwale, Ph.D

Academic Editor

PLOS One

Additional Editor Comments (optional):

Reviewers' comments:

Reviewer's Responses to Questions

**Comments to the Author**

Reviewer #3: All comments have been addressed

2. Is the manuscript technically sound, and do the data support the conclusions?

Reviewer #3: Yes

3. Has the statistical analysis been performed appropriately and rigorously?

Reviewer #3: Yes

4. Have the authors made all data underlying the findings in their manuscript fully available?

Reviewer #3: Yes

5. Is the manuscript presented in an intelligible fashion and written in standard English?

Reviewer #3: Yes

Reviewer #3: The article is well written, and the authors have addressed all reviewers' comments. However, there is a typo in the study design figure (Figure 1), Clinical revelence instead of clinical relevance.

**Do you want your identity to be public for this peer review?** For information about this choice, including consent withdrawal, please see our Privacy Policy

Reviewer #3: **Yes:** Mubarak Mojoyinola

---

## [Editor Report · Acceptance letter]

PONE-D-25-36938R1

PLOS One

Dear Dr. Nguyen,

I'm pleased to inform you that your manuscript has been deemed suitable for publication in PLOS One. Congratulations! Your manuscript is now being handed over to our production team.

Kind regards,

on behalf of

Dr Musa Adekunle Ayanwale

Academic Editor

PLOS One